# The Development of ICT-Based Exercise Rehabilitation Service Contents for Patients with Musculoskeletal Disorders and Stroke

**DOI:** 10.3390/ijerph19095022

**Published:** 2022-04-20

**Authors:** Jiyoun Kim, Jiyeon Song, Donguk Kim, Jinho Park

**Affiliations:** 1Department of Exercise Rehabilitation & Welfare, Gachon University, Incheon 21936, Korea; eve14jiyoun@gachon.ac.kr; 2Exercise Rehabilitation Convergence Institute, Gachon University, Incheon 21936, Korea; jean93sjy@gmail.com (J.S.); kim24607@gmail.com (D.K.); 3Gachon Biomedical Convergence Institute, Gachon University Gil Medical Center, Incheon 21565, Korea

**Keywords:** musculoskeletal disorders, stroke, personalized exercise rehabilitation program service, information and communication technology, hospital-community linked

## Abstract

Exercise rehabilitation services connecting hospitals and communities increase patient participation and improve quality of life by reducing medical expenses. South Korea’s multi-ministerial governments have been working together to develop ICT-based hospital-community-linked services to create an exercise program that the public can easily use. This study aims to develop the exercise rehabilitation service components for the application and prescription of ICT-based exercise programs implemented in hospitals and communities. A literature review was conducted, and an expert committee was comprised to classify the components of exercise rehabilitation services. As a result, we classified the first components as functional classification, rehabilitation area, equipment uses, exercise type, frequency, and intensity. Subsequently, exercise programs were developed by applying the first components. Based on the purpose of exercise rehabilitation, we classified the representative standard exercise and grouped the same exercise movements using tools and exercise machines. The finding of this study will help to give the correct exercise prescription and manage patients’ improvement process for exercise instructors. In addition, it guides patients in need of exercise rehabilitation to participate in an accurate and safe exercise in the community. This study is a novel attempt to develop ICT based hospital-community-linked exercise rehabilitation service for patients.

## 1. Introduction

According to the Korean Healthcare Big Data Hub, Alzheimer’s dementia was associated with the highest medical care costs, followed by stroke, knee arthritis, intervertebral disc disorder, shoulder lesions, and spinal diseases. The characteristics of these diseases can be divided into stroke and musculoskeletal disorders (MSD), with the exception of Alzheimer’s disease [1]. MSDs refer to disorders of the muscles, bones, and joints, such as the neck, shoulder, waist, and knee [2]. MSDs reduce overall physical function due to mobility limitations and pain during movement [2,3]. About 1.71 billion people worldwide have MSD, and it contributes greatly to the increase in years of living with disability (YLDs) [4]. In South Korea, about 17.61 million patients (34.3%) received treatment for MSD in 2019, and medical expenses were about US$66.256 billion, accounting for 10.9% of the total medical expenses of health insurance medical institutions [5].

Stroke is a disease caused by blocked or ruptured blood vessels that supply oxygen and nutrients to the brain and is a major risk factor for death and disability [6]. The number of stroke patients in Korea has increased from 93,670 in 2014 to approximately 120,000 in 2019, and this increase is expected to continue with the aging population [7]. The characteristics of these two diseases require constant rehabilitation, and when the timing of functional recovery is missed, it causes an increase in medical expenditure due to secondary disability and hospital revisits [8,9,10,11,12]. Muscle-strengthening exercises are the most common and effective treatment for MSDs, such as arthritis and back pain, as they can alleviate muscle strength and mass reduction, improve bone density, and improve physical function in rehabilitation patients [13]. Rodriguez et al. [14] demonstrated that patients with MSD can reduce shoulder, wrist, and spinal pain when performing muscle strengthening exercises with 70–85% of one repetition maximum (RM) three times a week for 20 min per session.

For stroke patients, it is important to improve balance and walking ability during early exercise rehabilitation, which is performed from the acute to sub-acute phases. An and Shaughnessy [15] suggested that balance exercises should be performed 3–5 times a week for at least 1 h per session, and walking exercises should be performed 3–5 times a week for 30 min per session. A systematic review reported that post-stroke strength training can improve functional activity, as measured by self-assessment measures such as maximal activity score (MAS) and human activity profile (HAP) [16]. Pang et al. [17] demonstrated that aerobic exercise of 3–5 days per week and 40–50% of heart rate reserve (HRR) for 20–40 min per session was effective in improving aerobic fitness, walking speed, and endurance in patients with mild and moderate stroke.

However, although many previous studies have demonstrated the effectiveness of exercise rehabilitation in patients with MSD and stroke, there is no consistent opinion on the duration of exercise programs by body part [18,19].

The rehabilitation medical system in the United States, Japan, and Australia is divided into three stages: acute (approximately 2 months), sub-acute (2–6 months), and chronic (6 months). Intensive and community-linked rehabilitation are provided during the postoperative recovery period and the maintenance and promotion period, respectively. In the United States, services tailored to patient needs are provided to help patients return to society [20]. However, although acute rehabilitation treatment is performed in hospitals in South Korea, functional rehabilitation is discontinued because there is no community-linked continuous rehabilitation exercise system available after discharge [20]. As a result, the number of patients who are not provided with systematic exercise rehabilitation programs is gradually increasing, delaying the patient’s return to daily life and causing a vicious cycle due to decreased physical function and increased medical expenditure due to revisiting hospitals [10,11,12].

To address this problem, the Ministry of Health and Welfare, Ministry of Science and Information Communications Technology (ICT), and Ministry of Culture, Sports, and Tourism have been working together to promote national research and development since 2021 (Figure 1). The Ministry of Health and Welfare and the Ministry of Science and ICT are in charge of establishing rehabilitation systems in hospitals, while the Ministry of Culture, Sports, and Tourism is in charge of establishing services to implement exercise rehabilitation programs for community patients after discharge (Ministry of Culture, Sports and Tourism, Sports Industry Technology Development Project No.1375027368). The developed community exercise rehabilitation service implements ICT and is customized to include exercise prescriptions, exercise programs, and patients’ use of exercise rehabilitation.

Therefore, this study was conducted to develop the components of the exercise rehabilitation service for the application and prescription of ICT-based exercise programs implemented in community exercise rehabilitation services. This study aims to develop accurate exercise prescriptions and exercise selection methods linked to hospital data. It also systematizes the composition that can be applied to ICT to develop exercise rehabilitation program services available in the exercise rehabilitation environment in South Korea.

## 2. Materials and Methods

### 2.1. Intervention Design and Participants

This study attempted to prepare a periodically sustainable exercise rehabilitation service system that integrates and manages medical institutions and communities composed of domestic pan-ministerial research (Ministry of Health and Welfare and Ministry of Culture, Sports, and Science) (Figure 1). First, exercise rehabilitation based on focus group interviews (FGI) is used to visualize the motor rehabilitation service elements and content components that are in line with our goal. The organization consisted of medical institutions and specialists with expertise in community movement rehabilitation. When recruiting focus groups in social science research, mixed groups are used more effectively to gain diverse perspectives and experiences than homogeneous groups [21,22].

In this study, we considered two factors in the recruitment process to obtain rich and accurate data. The first factor is the professional element, and the second factor is the career period. The professional element consisted of a total of 40 people, including medical specialists, athletic specialists, and ICT developers. The medical specialists consist of 13 people, including rehabilitation medicine, orthopedics, neurosurgeons, and physiotherapy specialists. Athlete specialists were composed of 19 people, including professors in the sports, sports medicine, exercise rehabilitation, and physical therapy field, researchers, and an exercise revitalization specialist instructor and a health exercise manager who have more than 5 years of experience in the field (Table 1). Finally, eight ICT developers from the Electronics and Telecommunications Research Institute (ETRI), who have more than five years of research experience, participated and played a role in embodying the developed exercise rehabilitation service component as ICT. Selected experts have agreed to participate in the development of community movement rehabilitation services through long-term, multidisciplinary, and personalized interventions (Figure 2).

### 2.2. Process

The procress of this research is as follows. To design a focus group guide, we first conducted a literature search on domestic and overseas exercise rehabilitation. FGI questions were based on the contents discovered during this process and confirmed information regarding the selection of the patient group requiring continuous rehabilitation, the purpose of exercise rehabilitation, the elements of exercise rehabilitation, and the criteria of exercise prescription. FGI was conducted through one-on-one and small group interviews, which focus on coordinating interactions between interviewees and require the organizers’ participation only in necessary circumstances. This method of research can collect comments from respondents through the interaction of the experts interviewed (Oh and So, 2022). In this study, FGI was conducted a total of 12 times to classify primary and secondary components. Eight interviews were conducted on extracting the primary components (session 1—target selection and rehabilitation site, sessions 2 and 3—motor function classification, session 4—tool utilization, session 5—exercise prescription (FITT), and sessions 6 to 8—ICT application systematization). Classification of rehabilitation exercises were extracted as secondary components from sessions 9 to 12 (Figure 2).

### 2.3. Theoretical Approach

It is important to accurately understand the human body’s movements to perform effective exercise rehabilitation. The motion system of the human body consists of muscular, skeletal, and nervous systems. Human movements consist of each system with its sub-components working interdependently [23]. If any components are damaged or impaired, movement may be restricted. Therefore, after accurately identifying the system for movement, exercise prescriptions should be made in consideration of the characteristics of each joint [24]. For functional classification, it is necessary to understand the three planes (frontal, transverse, and sagittal planes) in which the movement of the joints occurs (Table 2) [25].

Exercise prescriptions and programs are implemented based on frequency, intensity, time, type, volume, and progression (FITT-VP) from the American College of Sports Medicine (ACSM) Guidelines for Exercise Testing and Prescription [26], a representative guideline for patient exercise prescription. In addition, according to exercise guidelines from the ACSM and the Centers for Disease Control and Prevention, adults aged 18–65 years participated in at least 150 min of moderate-intensity aerobic physical activity and 2 days of weight training per week, which should be applied when developing exercise rehabilitation programs [26].

In this study, based on these theories, the movements of the exercise rehabilitation service were reclassified based on seven functional movements and five combined joint movements in the functional classification system of movement. In addition, for the accuracy of exercise prescription, exercise frequency, intensity, time, and type based on the FITT-VP theory of ACSM were selected as program components for exercise rehabilitation services.

## 3. Results

### 3.1. First Components Selection of the Exercise Rehabilitation Service

In this study, the components of the exercise rehabilitation service were extracted with the first and second process through theoretical approach and discussions by the expert committee. The primary components have extracted functional classification, rehabilitation area, equipment use, exercise type, frequency, and intensity (Table 3).

(1)Selection of the subject patients group

The target patient group of the exercise rehabilitation program developed in this study was set as MSD and stroke patients who needed continuous application of the exercise rehabilitation program in the community after hospital treatment. Patients with MSD include those with shoulder joint injury, degenerative arthritis, and neck and vertebral disc disease. With the doctor’s prescription, stroke patients included those with cerebral infarction, and are selected as mild patients (walkable) living in the community (those who could provide exercise rehabilitation services). The detailed target group is summarized in the patient group in Table 3. The body was classified into three parts (upper extremity, lower extremity, and trunk) and the whole body and one representative disease that frequently occurs in each area was set as the disease subject to exercise rehabilitation.

(2)Functional movement classification

Functional classification of the movement of human joints was performed. Representative movements occurring in each joint were classified into kinematic terms and 9 functional movements [27].

(3)Use of tools

The exercise rehabilitation service developed in this study can be used as a personalized program for hospitals and community settings. In other words, users who use the program are intended to participate in exercise rehabilitation services suitable for situations in various places, such as hospitals, exercise centers, and homes. Therefore, to enable exercise rehabilitation in various places, various exercise tools, including bodyweight exercise without tools, exercise using various small tools such as dumbbells and bands, and exercise using various machines, are set as elements to constitute exercise rehabilitation services suitable for different environments. Table 4 lists the equipment used in this study.

(4)Type of exercise

Patients in need of rehabilitation have different exercise methods depending on the degree of rehabilitation, and it is necessary to appropriately change and implement exercise type according to purpose. In this study, health-related physical fitness elements (cardiopulmonary endurance, muscle endurance, flexibility) and skill-related physical fitness elements (agility, coordination, equilibrium), as defined in the ACSM [26], were set as service components.

(5)Frequency and intensity

In terms of exercise frequency, it is necessary to gradually increase the frequency and intensity according to the patient’s rehabilitation progress. In particular, stroke patients with cardiovascular disease should set a low frequency and intensity of exercise due to decreased physical ability caused by brain damage in the acute phase and gradually increase through the sub-acute and recovery periods [28]. Accordingly, the exercise frequency was set at a daily, weekly, and monthly basis. The intensity of exercise was selected in three stages: low-intensity exercise for early rehabilitation users or elderly users and moderate-intensity exercise for patients aiming to restore and maintain physical ability.

### 3.2. Second Components Selection of the Exercise Rehabilitation Service

Exercise movements were constructed by referring to various studies and books on the composition of exercise movements in the exercise rehabilitation service. Examples include exercise therapy books, therapeutic exercise [29], strength training anatomy [30], and Pilates bodies [31]. The model of the predictive exercise rehabilitation service by selecting the above components is as follows (Figure 3).

### 3.3. Differentiation of Exercise Rehabilitation Service Composition—1: N Movement Classification

The primary components of each exercise set were selected and classified. The classification order was as follows: first, the exercise rehabilitation contents were classified by rehabilitation part and anatomical movement function of the human body. Second, it is necessary to present exercise movements in the service differently, such as in the presence or absence of equipment, depending on the frequency, intensity, and type of exercise. Therefore, it is necessary to systematically classify the exercise movements used in exercise rehabilitation programs rather than indiscriminately listing them. The exercise rehabilitation was specified as a 1: N exercise movements classification in Table 4. In this study, we classified the representative standard exercise and grouped the same exercise movements including tools and exercise machines (Table 4).

### 3.4. Monitoring Plan

The effectiveness verification of the developed program in 2022 is planned to proceed as follows. First, the Ministry of Science and ICT is developing an exercise rehabilitation functional code (ERFC). This system can sufficiently convey therapeutic information about existing patients when patients in need of exercise rehabilitation have difficulty carrying out continuous rehabilitation in the community. It evaluates clinical function test elements such as walking, joint range of motion, stiffness, and pain, and evaluates physical strength factors such as muscle strength, cardiopulmonary endurance, and flexibility, and functional evaluation factors such as daily living ability, balance, and systemic function. In this study, the code will be issued to the patient, and the exercise service program developed in this study will be conducted for 3 to 6 months. Effectiveness verification will repeatedly measure the above evaluation factors of ERFC and verify the program’s effectiveness in exercise rehabilitation services. In addition, the feasibility of the exercise rehabilitation program was verified using expert opinions (Figure 4). The order of the validity verification was as follows: first, after selecting a specialized panel, such as a health exercise manager, the program will conduct a feasibility study and then repeat the process of collecting and analyzing expert opinions to finally synthesize opinions to derive feasibility predictions. Finally, we plan to survey program users’ satisfaction with the program’s difficulty suitability and exercise stimulation levels (Figure 4). Based on this evaluation, we plan to pursue the advancement of the functional exercise rehabilitation service model based on the ICT classification.

After verifying service validity and program effectiveness, patients’ accumulated program usage data will be tested for T-test, one-way analysis of variance (ANOVA), and two-way ANOVA using SPSS to determine if there were statistically significant results.

## 4. Discussion

This study was conducted to extract and systematize components for developing personalized ICT-based exercise rehabilitation programs for use in hospitals and communities for patients with MSDs and strokes.

Many ICT-based exercise programs have been developed. For example, cycling pelotons are an ICT-based exercise program that combines the visual sensory effect of playing with others through internet access when cycling indoors [32]. In addition, the Internet of Things-based smart mirror is an example of applying an ICT exercise program to the mirror to proceed with individualized personal exercise at home and is designed to promote the health of users [33]. However, this has the disadvantage of providing exercise programs to maintain and promote the general public’s health, making it difficult to use for patients in need of rehabilitation, and may pose a risk of injury.

The number of patients with MSDs and strokes in South Korea is increasing every year, and as a result, medical expenses are also increasing [1]. In addition, due to the lack of specialized exercise rehabilitation facilities in the community and high medical costs, patient management is insufficient until complete recovery is achieved immediately after discharge [24,34]. As for community rehabilitation services available in South Korea, there are 76 fitness certification centers nationwide called the National Fitness 100, installed by each region community [35]. The National Fitness 100 is a state-designated public certification institution that measures, evaluates people’s fitness levels, and scientifically provides exercise counseling and prescriptions. The institution provides customized exercise programs suitable for physical fitness levels based on the measurement results so that participants can consistently participate in sports to improve physical fitness in their daily lives and issue a national certificate for physical fitness [36]. However, since the fitness measurement methods and indicators suggested by the National Fitness 100 are set based on the general public, not rehabilitation patients, it is difficult for patients with MSDs and strokes to participate in the measurement and accurate interpretation of results [37]. Therefore, most patients who need exercise rehabilitation have no choice but to use private exercise rehabilitation centers, which leads to a high burden of medical costs, increasing the rehabilitation rate before full recovery [38].

The ICT-based exercise rehabilitation service developed in this study is applied in consideration of the physical fitness level of patients in need of exercise rehabilitation and consists of small group exercise and training using machines to use the environmental infrastructure of the community. In other words, it is composed of the goal of implementing a service that enables hospitals and local communities to effectively restore physical function and promote participation in daily life through linked exercise rehabilitation programs.

With the onset of the fourth industrial revolution, various ICT-applied exercise services have the advantage of improving motivation for participation in exercise so that participants can lead a healthy life and manage health easily and conveniently [30,36]. In addition, since these services enable personalized exercise prescriptions through accumulated exercise data collection and analysis, they consequently increase the persistence and effectiveness of exercise and help maintain healthy lifestyles [39].

Even within hospitals, the use of exercise rehabilitation equipment using ICT is increasing. For example, Shin et al. [38] applied a task-specific interactive game-based virtual reality rehabilitation system in hospitals for stroke patients. They found that it was a safe and suitable rehabilitation tool for improving motor function among patients with various levels of disease. In addition, there was no significant difference in the effects of movement rehabilitation between the augmented reality (AR) and non-AR exercise groups for patients undergoing lower extremity rehabilitation exercise [40]. However, there has been no previous research in South Korea on the implementation of exercise rehabilitation program services linked to hospitals and local communities.

This study aimed to implement exercise rehabilitation program services within the community. The first purpose of this study was to give the correct exercise prescription and manage patients’ improvement process for exercise instructors. Second, it guides patients in need of exercise rehabilitation to participate in accurate and safe exercise rehabilitation in the community. According to a study by Palazzo et al. [41], stroke patients cannot complete their exercise rehabilitation because of a lack of social support and feedback. When an exercise rehabilitation program service is implemented, patients who participate in the program are expected to promote their return to daily life by understanding their condition and participating in systematic exercise rehabilitation programs.

When prescribing exercise, the characteristics of the disease, clarity of rehabilitation areas, and systematic goals for improving motor function are important [26]. In other words, it is crucial whether the exercise intensity and frequency of the configured program can be appropriately applied to individuals in accurately understanding the patient’s functional state and applying the exercise program. This is because it is an important factor in the rehabilitation process of functional recovery.

Based on these previous studies, this study proposed the components and systematization of ICT-applicable exercise rehabilitation programs linked to hospitals in the local community. The developed exercise program is planned for a pilot project linking hospitals and communities by 2023. Its effectiveness will be verified through pilot tests in 2022, and further development will be made through feedback. All processes, such as applicant selection, functional stage development, and program organization, were verified and implemented by the expert committee. Through this study, it is expected that patients requiring rehabilitation will be able to participate in community-based exercise rehabilitation programs until they recover to their normal life pattern, and medical expenses will be reduced. The connection with hospitals and local communities will ultimately maintain and improve patients’ health and their quality of life. The limitation of this study is that there is no clear standard for the ICT-based exercise program. There are difficulties in recruiting subjects to receive the exercise rehabilitation program, the lack of prior research that can be referred to as the first study in Korea, and the fact that the program has not been verified to meet the purpose of this study. Efficiency of ICT-based programs and infrastructure construction are important elements for a successful outcome. Nevertheless, since many previous studies [13,14,15] show that continuous exercise rehabilitation in hospital and community settings has a positive effect on patients with MSDs and strokes, this study can be seen as meaningful.

## 5. Conclusions

This study is a novel attempt to develop the components of an exercise rehabilitation program service for the application and prescription of ICT-based exercise programs implemented in hospital-community-linked services. Consequently, it is expected that services can be implemented by extracting and systematizing appropriate components in implementing exercise rehabilitation programs. The project is scheduled to be implemented between 2021 and 2023. So far, 37 expert committees consisting of 13 hospital medical specialists, 19 exercise rehabilitation experts, and five ICT program implementation experts have discussed extracting the elements necessary for the exercise rehabilitation program. Exercise movements were classified according to ACSM guidelines as a method for developing more accurate exercise programs. In addition, we have developed an exercise rehabilitation service by systematically classifying the exercise rehabilitation programs currently being implemented in hospitals and rehabilitation facilities according to patient level. It is planned that this service will be implemented in the community in connection with hospitals in early 2023.

## Figures and Tables

**Figure 1 ijerph-19-05022-f001:**
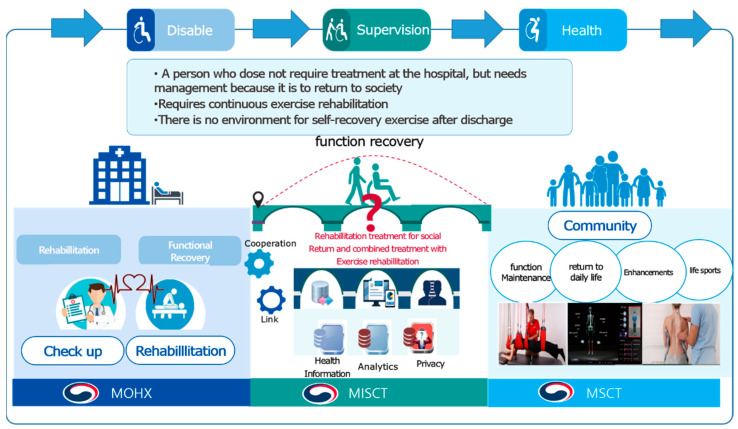
Inter-ministerial collaboration model. MOHW: Ministry of Health and Welfare, MISCT: Ministry of Science and ICT, MCST: Ministry of Culture, Sport and Tourism.

**Figure 2 ijerph-19-05022-f002:**
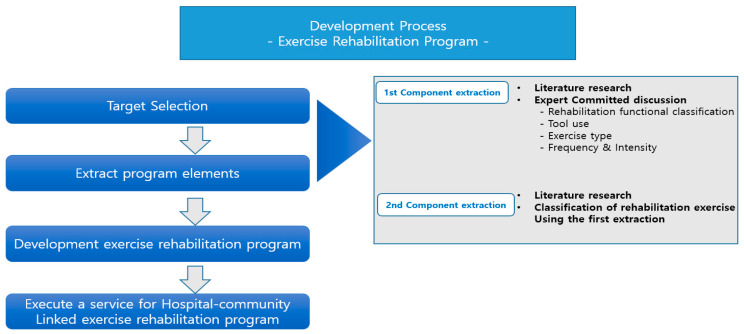
The whole development process—exercise rehabilitation program.

**Figure 3 ijerph-19-05022-f003:**
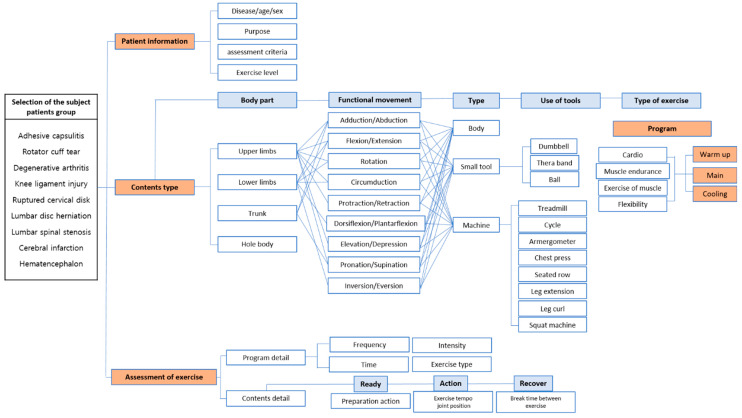
Exercise rehabilitation service model map.

**Figure 4 ijerph-19-05022-f004:**
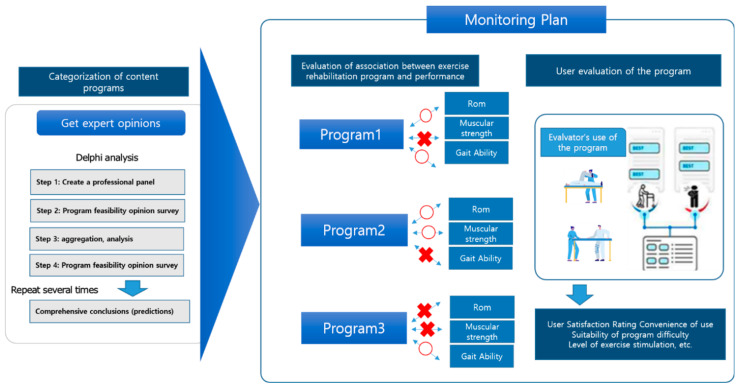
Monitoring plan.

**Table 1 ijerph-19-05022-t001:** Focus group interview (FGI) characteristics.

Name	Specialists Classification	Institution	Career/ Year	Name	Specialists Classification	Institution	Career/ Year
S1	Medical Specialists	Doctor	10	S21	Exercise Rehabilitation Specialists	Researcher	5
S2			16	S22			9
S3			7	S23			6
S4			5	S24			8
S5			8	S25			8
S6			7	S26		Exercise Specialist	7
S7		Physical Therapist	21	S27			9
S8			16	S28			10
S9			8	S29			6
S10			9	S30			12
S11			7	S31			5
S12			11	S32			8
S13			6	S33	ICT Development Specialist	Electronics and Telecommunications Research Institute	8
S14	Exercise Rehabilitation Specialists	Professor	7	S34			10
S15			7	S35			7
S16			14	S36			6
S17			11	S37			16
S18			12	S38			14
S19			6	S39			9
S20			8	S40			12

**Table 2 ijerph-19-05022-t002:** Classification of joint movements by movement plane.

Plane	Joint Movements
Sagittal plane	Flexion/Extension
Frontal plane	Abduction/Adduction
	Lateral Flexion
	Inversion/Eversion
Transverse plane	Lateral Rotation/External Rotation
	Spinal Rotation
	Horizontal Adduction/Horizontal Abduction
Combined joint movement	Protraction/Retraction
	Dorsiflexion/Plantar flexion
	Elevation/Depression
	Pronation/Supination
	Circumduction

**Table 3 ijerph-19-05022-t003:** Results of ICT-based exercise rehabilitation services component extraction and systematization of scenarios.

Category	Theme Cluster	Theme	
Selection of the subject patients group	Adhesive capsulitis	Shoulder Joint	Upper Limbs
	Rotator cuff tear		
	Degenerative arthritis	Knee Joint	Low Limbs
	Knee ligament injury		
	Ruptured cervical disk	Cervical	Trunk
	Lumbar disc herniation	Lumbar	
	Lumbar spinal stenosis		
	Cerebral infarction	Brain-Blood Vessel	Whole Body
	Hematencephalon		
Functional movement classification	Adduction/Abduction	Adduction/Abduction	
		Horizontal Adduction/Horizontal Abduction	
	Flexion/Extension	Flexion/Extension	
		Lateral Flexion	
	Rotation	Lateral Rotation/External Rotation	
		Spinal Rotation	
	Circumduction	Circumduction	
	Protraction/Retraction	Protraction/Retraction	
	Dorsiflexion/Plantarflexion	Dorsiflexion/Plantarflexion	
	Elevation/Depression	Elevation/Depression	
	Pronation/Supination	Pronation/Supination	
	Inversion/Eversion	Inversion/Eversion	
Use of tools	Small Tool	Chair, Dumbbell, Thera band, Ball, Step Box	
	Machine	Treadmill, Cycle, Arm Ergometer, Chest Press, Seated Row, Butterfly, Leg Press, Leg Extension, Leg Curl, Squat Machine, Sit Up, Back Extension	
Type of exercise	Health-related physical fitness elements	Cardiopulmonary Endurance, Muscle Endurance, Flexibility	
	Skill-related physical fitness elements	Agility, Coordination, Balance	

**Table 4 ijerph-19-05022-t004:** Examples of 1: N movement classification.

Muscular Strength						
Body Part	Category	Standard Exercise (1)	Category		Exercise Variations (N)	
Upper Limbs	Picture	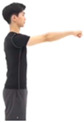	Picture	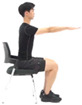	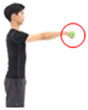	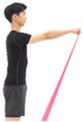
			Name	Shoulder flexion (seated chair, one arm)	Shoulder flexion (using dumbbell, one arm)	Shoulder flexion (using band, one arm)
			Equipment	Chair	Dumbbells	Band
			Frequency	2–3 days/week	2–3 days/week	2–3 days/week
			Intensity	40–60% of MVC	40–60% of MVC	40–60% of MVC
			Volume	Repetitions: 10	Repetitions: 10	Repetitions: 10
				Sets: 3–4	Sets: 3–4	Sets: 3–4
				Pattern: 1 min	Pattern: 1 min	Pattern: 1 min
			Picture	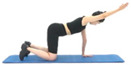	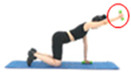	
	Name	Shoulder flexion (standing, one arm)	Name	Shoulder flexion (prone position, one arm)	Shoulder flexion (prone position, using dumbbell, one arm)	
	Use of tools	None	Use of tools	None	Dumbbells	
	Frequency	2–3 days/week	Frequency	2–3 days/week	2–3 days/week	
	Intensity	40–60% of MVC	Intensity	40–60% of MVC	40–60% of MVC	
	Volume	Repetitions: 10	Volume	Repetitions: 10	Repetitions: 10	
		Sets: 3–4		Sets: 3–4	Sets: 3–4	
		Pattern *: 1 min		Pattern: 1 min	Pattern: 1 min	

MVC: maximal voluntary contraction, * Pattern denotes rest time.

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
