# Peer review of "The Development of ICT-Based Exercise Rehabilitation Service Contents for Patients with Musculoskeletal Disorders and Stroke"

_ijerph, 2022, doi:10.3390/ijerph19095022_

Round 1

Reviewer 1 Report

First of all, the English language must be refined!

Secondly the abstract must be redone!

In the introduction section, the authors begin the paper on musculoskeletal disorders…I see no connection, and afterwards, they make statements regarding post-stroke rehabilitation. If the authors are considering their research in the field of rehabilitation in post-stroke survivors, they should only address this issue!

The introduction starts from the incidence and prevalence of diseases that cause disability and consequently consume medical services, thus gathering diseases of various categories such as Alzheimer's, Herniated Disc, Stroke, Knee Arthritis or Shoulder Injury. Then it goes to stroke and motivates the importance of rehabilitation after stroke. There are general claims about muscle strength exercises in general, which are said to be the most common and effective treatment for MSDs, such as arthritis and back pain as well as stroke.

I don't think it's fair to gather all these diseases together and especially to make general statements about the exercise good for all. It is scientifically correct to customize the rehabilitation program for each disease, and then to individualize the rehabilitation program for each patient.

For example, in lines 44-53 the authors refer to straightening exercises being good for both MSD and post-stroke rehabilitation, but do not take into consideration spasticity! Therefore, the research is not appropriate!

Lines 85-87- are the authors describing what will happen!? It is not clear!

The methodology: Things are very homogeneous and there is no concrete classification/quantification! In addition, if the authors really wanted to be based on the opinion of "experts" in the field, questionnaires had to be made with the triangulation of the data obtained to determine some objective opinions.

In the results, and also in the discussions, the authors relate to rather subjective aspects… I did not see any objective evaluation… It is as if any experienced physiotherapist would give their opinion about what is good to do with patients…

The paper/the research needs many serious changes and objective quantifiable assessment methods!

Author Response

Thank you for reviewing our manuscript, and we appreciate your comments. We have revised the limitations in accordance with your recommendations and highlighted them in yellow.

First of all, the English language must be refined!

Thank you for the suggestion. Overall refinement has been made to the manuscript's English language.

Secondly the abstract must be redone!

Thank you for the suggestion. We have revised the abstract. Please see below.

Lines 11-26.

Exercise rehabilitation services connecting hospitals and communities increase patients' participation and improve the quality of life by reducing medical expenses. South Korea's multi-ministerial governments have been working together to develop ICT-based hospital-community-linked services to create an exercise program that the public can easily use. This study aims to develop the exercise rehabilitation service components for the application and prescription of ICT-based exercise programs implemented in hospitals and communities. A literature review was conducted, and an expert committee was comprised to classify the components of exercise rehabilitation services. As a result, we classified the primary components as functional classification, rehabilitation area, equipment uses, exercise type, frequency, and intensity. Subsequently, exercise programs were developed by applying the primary components. Based on the purpose of exercise rehabilitation, we classified the representative standard exercise and grouped the same exercise movements using tools and exercise machines. The finding of this study will help to give the correct exercise prescription and manage patients' improvement process for exercise instructors. In addition, it guides patients in need of exercise rehabilitation to participate in an accurate and safe exercise in the community. This study is a novel attempt to develop ICT based hospital-community-linked exercise rehabilitation service for patients.

In the introduction section, the authors begin the paper on musculoskeletal disorders…I see no connection, and afterwards, they make statements regarding post-stroke rehabilitation. If the authors are considering their research in the field of rehabilitation in post-stroke survivors, they should only address this issue!

The purpose of this study is to create individual exercise rehabilitation programs related to stroke and musculoskeletal disorders for patients in need of reduced medical costs and exercise rehabilitation.

We revised lines 31-35.

According to the Korean Healthcare Big Data Hub, Alzheimer's dementia was associated with the highest medical care costs, followed by Stroke, knee arthritis, intervertebral disc disorder, shoulder lesions, and spinal diseases. The characteristics of these diseases can be divided into stroke and Musculoskeletal disorders (MSD), with the exception of Alzheimer's disease [1].

The introduction starts from the incidence and prevalence of diseases that cause disability and consequently consume medical services, thus gathering diseases of various categories such as Alzheimer's, Herniated Disc, Stroke, Knee Arthritis or Shoulder Injury. Then it goes to stroke and motivates the importance of rehabilitation after stroke. There are general claims about muscle strength exercises in general, which are said to be the most common and effective treatment for MSDs, such as arthritis and back pain as well as stroke.

I don't think it's fair to gather all these diseases together and especially to make general statements about the exercise good for all. It is scientifically correct to customize the rehabilitation program for each disease, and then to individualize the rehabilitation program for each patient.

For example, in lines 44-53 the authors refer to straightening exercises being good for both MSD and post-stroke rehabilitation, but do not take into consideration spasticity! Therefore, the research is not appropriate!

Thank you for the question. Depending on our research objectives, we have distinguished between the two diseases (Stroke and MSDs) and added the missing studies.

We revised lines 59-61.

A systematic review reported that post-stroke strength training can improve functional activity, as measured by self-assessment measures such as maximal activity score (MAS) and human activity profile (HAP) [16].

Lines 85-87- are the authors describing what will happen!? It is not clear!

We revised lines 87-90.

The developed community exercise rehabilitation service implements ICT and is customized to include exercise prescriptions, exercise programs, and patients' use of exercise rehabilitation.

The methodology: Things are very homogeneous and there is no concrete classification/quantification! In addition, if the authors really wanted to be based on the opinion of "experts" in the field, questionnaires had to be made with the triangulation of the data obtained to determine some objective opinions.

In the results, and also in the discussions, the authors relate to rather subjective aspects… I did not see any objective evaluation… It is as if any experienced physiotherapist would give their opinion about what is good to do with patients…

We revised lines 102-125.

This study attempted to prepare a periodically sustainable exercise rehabilitation service system that integrates and manages medical institutions and communities composed of domestic pan-ministerial research (Ministry of Health and Welfare and Ministry of Culture, Sports, and Science) (Figure 1. First, exercise rehabilitation Based on FGI (focus group interviews), it is to visualize the motor rehabilitation service elements and content components that are in line with our goal. The organization consisted of medical institutions and specialists with expertise in community movement rehabilitation. When recruiting focus groups in social science research, mixed groups are used more effectively to gain diverse perspectives and experiences than homogeneous groups (21-22).

In this study, we considered two factors in the recruitment process to obtain rich and accurate data. The first factor is the professional element, and the second factor is the career period. The professional element consisted of a total of 40 people, including medical specialists, athletic specialists, and ICT developers. The medical specialists consist of 13 people, including rehabilitation medicine, orthopedics, neurosurgeons, and physiotherapy specialists. Athlete specialists were composed of 19 people, including professors in the sports, sports medicine, exercise rehabilitation, and physical therapy field, researchers, and an exercise revitalization specialist instructor and a health exercise manager who have more than 5 years of experience in the field (Table 1). Finally, 8 ICT developers from the Electronics and Telecommunications Research Institute (ETRI), who have more than five years of research experience, participated and played a role in embodying the developed exercise rehabilitation service component as ICT. Selected experts have agreed to participate in the development of community movement rehabilitation services through long-term, multidisciplinary and personalized interventions. (Figure 2).

We revised lines 131-145.

The progress of this research is as follows. To design a focus group guide, we first conducted a literature search on domestic and overseas exercise rehabilitation. FGI questions were based on the contents discovered during this process and confirmed information regarding the selection of the patient group requiring continuous rehabilitation, the purpose of exercise rehabilitation, the elements of exercise rehabilitation, and the criteria of exercise prescription. FGI was conducted through one-on-one and small group interviews, which focus on coordinating interactions between interviewees and require the organizers' participation only in necessary circumstances. This method of research can collect comments from respondents through the interaction of the experts interviewed (Oh and So, 2022). In this study, FGI was conducted a total of 12 times to classify primary and secondary components. Eight interviews were conducted on extracting the primary components (session 1 - target selection and rehabilitation site, sessions 2 and 3 - motor function classification, session 4 - tool utilization, session 5 - exercise prescription (FITT), and sessions 6 to 8 -ICT application systematization). Classification of rehabilitation exercises were extracted as secondary components from sessions 9 to 12.

Also, Table 1 and Table 3 has been revised. Contents for 2.2 and 2.3 has been switched. Results 3.1, 3.2, and 3.3 has been revised.

The paper/the research needs many serious changes and objective quantifiable assessment methods!

Thank you for your comments. Overall revision has been made. Please review.

Reviewer 2 Report

Respected authors,

thank you for this interesting manuscript. I have some suggestions.

The objective is wide and unprecise.  I suggest you to writte the main objective and additional objectives.

The Metod: The sample is too small. How was the sample chosen? What type of study have you done? When was the study carried ? The Number of Decission and the date of approval are needed in the Method.

The conclusion is written in general. It must be more precise.

Author Response

Thank you for reviewing our manuscript, and we appreciate your comments. We have revised the limitations in accordance with your recommendations and highlighted them in yellow.

Thank you for this interesting manuscript. I have some suggestions.

The objective is wide and unprecise. I suggest you to writte the main objective and additional objectives.

Thank you for the suggestion. Depending on our research objectives, we have distinguished between the two diseases (Stroke and MSDs) and added the missing studies.

We revised lines 59-61.

A systematic review reported that post-stroke strength training can improve functional activity, as measured by self-assessment measures such as maximal activity score (MAS) and human activity profile (HAP) [16].

We revised lines 87-90.

The developed community exercise rehabilitation service implements ICT and is customized to include exercise prescriptions, exercise programs, and patients' use of exercise rehabilitation.

The Method: The sample is too small. How was the sample chosen? What type of study have you done? When was the study carried ? The Number of Decission and the date of approval are needed in the Method.

We revised lines 102-125.

 This study attempted to prepare a periodically sustainable exercise rehabilitation service system that integrates and manages medical institutions and communities composed of domestic pan-ministerial research (Ministry of Health and Welfare and Ministry of Culture, Sports, and Science) (Figure 1. First, exercise rehabilitation Based on FGI (focus group interviews), it is to visualize the motor rehabilitation service elements and content components that are in line with our goal. The organization consisted of medical institutions and specialists with expertise in community movement rehabilitation. When recruiting focus groups in social science research, mixed groups are used more effectively to gain diverse perspectives and experiences than homogeneous groups [21-22].

In this study, we considered two factors in the recruitment process to obtain rich and accurate data. The first factor is the professional element, and the second factor is the career period. The professional element consisted of a total of 40 people, including medical specialists, athletic specialists, and ICT developers. The medical specialists consist of 13 people, including rehabilitation medicine, orthopedics, neurosurgeons, and physiotherapy specialists. Athlete specialists were composed of 19 people, including professors in the sports, sports medicine, exercise rehabilitation, and physical therapy field, researchers, and an exercise revitalization specialist instructor and a health exercise manager who have more than 5 years of experience in the field (Table 1). Finally, 8 ICT developers from the Electronics and Telecommunications Research Institute (ETRI), who have more than five years of research experience, participated and played a role in embodying the developed exercise rehabilitation service component as ICT. Selected experts have agreed to participate in the development of community movement rehabilitation services through long-term, multidisciplinary and personalized interventions. (Figure 2).

We revised lines 131-145.

The progress of this research is as follows. To design a focus group guide, we first conducted a literature search on domestic and overseas exercise rehabilitation. FGI questions were based on the contents discovered during this process and confirmed information regarding the selection of the patient group requiring continuous rehabilitation, the purpose of exercise rehabilitation, the elements of exercise rehabilitation, and the criteria of exercise prescription. FGI was conducted through one-on-one and small group interviews, which focus on coordinating interactions between interviewees and require the organizers' participation only in necessary circumstances. This method of research can collect comments from respondents through the interaction of the experts interviewed (Oh and So, 2022). In this study, FGI was conducted a total of 12 times to classify primary and secondary components. Eight interviews were conducted on extracting the primary components (session 1 - target selection and rehabilitation site, sessions 2 and 3 - motor function classification, session 4 - tool utilization, session 5 - exercise prescription (FITT), and sessions 6 to 8 -ICT application systematization). Classification of rehabilitation exercises were extracted as secondary components from sessions 9 to 12.

The conclusion is written in general. It must be more precise.

We added lines 377-381.

Exercise movements were classified according to ACSM guidelines as a method for developing more accurate exercise programs. In addition, we have developed an exercise rehabilitation service by systematically classifying the exercise rehabilitation programs currently being implemented in hospitals and rehabilitation facilities according to the patient level.

Reviewer 3 Report

I found this study very interesting.
Currently, many patients with musculoskeletal disorders and strokes do not receive adequate exercise rehabilitation services after discharge. As a result of this study, it is expected that the rehabilitation program linked with the hospital and the community will be developed.
I hope my comments will be helpful for the study.

2.3. Process section
Please describe in detail the procedure for selecting subjects for this study.
 (Selection and exclusion criteria; Were stroke patients with musculoskeletal problems an exclusion criterion?)

Line 240: Table 1. Examples of 1: N movement classification. -> Please change to Table 5. Examples of 1: N movement classification

4. Discussion section
If there are limitations and disadvantages of the ICT based exercise program, please describe and suggest directions for improvement in this study.

Author Response

Thank you for reviewing our manuscript, and we appreciate your comments. We have revised the limitations in accordance with your recommendations and highlighted them in yellow.

I found this study very interesting.

Currently, many patients with musculoskeletal disorders and strokes do not receive adequate exercise rehabilitation services after discharge. As a result of this study, it is expected that the rehabilitation program linked with the hospital and the community will be developed.

I hope my comments will be helpful for the study.

2.3. Process section

Please describe in detail the procedure for selecting subjects for this study.

(Selection and exclusion criteria; Were stroke patients with musculoskeletal problems an exclusion criterion?)

Thank you for the suggestion. The purpose of this study is to develop an exercise rehabilitation service for target patients using results from literature review and the expert committee. 

Lines 115-123.

The medical specialists consist of 13 people, including rehabilitation medicine, orthopedics, neurosurgeons, and physiotherapy specialists. Athlete specialists were composed of 19 people, including professors in the sports, sports medicine, exercise rehabilitation, and physical therapy field, researchers, and an exercise revitalization specialist instructor and a health exercise manager who have more than 5 years of experience in the field (Table 1). Finally, 8 ICT developers from the Electronics and Telecommunications Research Institute (ETRI), who have more than five years of research experience, participated and played a role in embodying the developed exercise rehabilitation service component as ICT.

Line 240: Table 1. Examples of 1: N movement classification. -> Please change to Table 5. Examples of 1: N movement classification

Table 5 indicated

If there are limitations and disadvantages of the ICT based exercise program, please describe and suggest directions for improvement in this study.

Lines 359-365.

The limitation of this study is that there is no clear standard for the ICT-based exercise program. There are difficulties in recruiting subjects to receive the exercise rehabilitation program, the lack of prior research that can be referred to as the first study in Korea, and the fact that the program has not been verified to meet the purpose of this study. Efficiency of ICT-based programs and infrastructure construction are important elements for a successful outcome.

Round 2

Reviewer 1 Report

I respect the high expertise in the field of all the specialists you are involved in the project. I do not dispute their training and professional experience. I understand that you have tried to achieve multidisciplinary collaboration and that the key to the success of the application of ICT would be to reduce the costs of rehabilitation. I welcome the participation of medical and government institutions.

I agree that using telemedicine means we can do this but in different stages of the pathology.

What bothers me and I can't agree is the generalization of the statements for all the pathologies. I cannot agree that post-stroke rehabilitation can be done in the community through ICT, based on a standard program applied uniformly.

You can do this on muscle pathologies, osteoarthritis, and sports injuries, but not for stroke.

You state

The purpose of this study is ” to create individual exercise rehabilitation programs related to stroke and musculoskeletal disorders for patients in need of reduced medical costs and exercise rehabilitation”.

And yet your entire build supports the standardization of post-stroke rehabilitation for all patients, standardization, and application by a number of movement and sports specialists. I do not agree with that. You can refer to prophylaxis and gather a stroke together with other pathologies. You can also exclude from your statements acute and subacute stroke, which cannot be rehabilitated in the community and through telemedicine. You state that you select only mild stroke (186-188), but this means to specify the limitations of the study that it applies only to mild cases. And it is not fair to make it clear that the exercises that apply are the same even if you have structured them on different intensities of stress.

I do not agree with the statements in lines 219-227: Patients with stroke and cardiovascular disease do not do cardio training in the acute phase, even gradually. They have hemiplegia and are fighting for their lives, they are passively mobilized, and that is correctly and ethically done in the specialized neurorehabilitation wards. The cardio rehabilitation program set forth in lines 158-165 does not apply.

I do not understand the terms: hematencephalon (in table 3, instead of hemorrhagic stroke I suppose, exercise rehabilitation service, and movement rehabilitation services, have nothing to do with medical rehabilitation.

So, my final suggestion is to completely separate the two categories, extremely different as pathophysiology, clinic and of course medical approach, principles and techniques of rehabilitation!!!!!

Author Response

Below is the comment from the reviewer and our response. The reviewer’s comments are in black, and our responses are in red.

Reviewer 1 Comment

I respect the high expertise in the field of all the specialists you are involved in the project. I do not dispute their training and professional experience. I understand that you have tried to achieve multidisciplinary collaboration and that the key to the success of the application of ICT would be to reduce the costs of rehabilitation. I welcome the participation of medical and government institutions.

I agree that using telemedicine means we can do this but in different stages of the pathology.

What bothers me and I can't agree is the generalization of the statements for all the pathologies. I cannot agree that post-stroke rehabilitation can be done in the community through ICT, based on a standard program applied uniformly.

You can do this on muscle pathologies, osteoarthritis, and sports injuries, but not for stroke.

You state

The purpose of this study is “ to create individual exercise rehabilitation programs related to stroke and musculoskeletal disorders for patients in need of reduced medical costs and exercise rehabilitation”.

And yet your entire build supports the standardization of post-stroke rehabilitation for all patients, standardization, and application by a number of movement and sports specialists. I do not agree with that. You can refer to prophylaxis and gather a stroke together with other pathologies. You can also exclude from your statements acute and subacute stroke, which cannot be rehabilitated in the community and through telemedicine. You state that you select only mild stroke (186-188), but this means to specify the limitations of the study that it applies only to mild cases.

First of all, thank you for your in-depth comments.

We are currently participating in a large national R&D project and have applied only a portion of the information in the research paper. This seems to have caused the possible misunderstanding.

(Line 186-188) The concept of “mild stroke” is a person who has a minor stroke of 5 or less in the NIHSS (National Institutes of Health Stroke Scale) evaluation score and applies only to a patient who needs exercise rehabilitation under the judgment of the doctor at the time of discharge.

There are many acute and sub-acute patients who need continuous exercise rehabilitation in the community, not in the hospital, based the National Institutes of Health Stroke Scale (NIHSS) standards. In addition, maintaining function and safety of gait is essential in exercise rehabilitation for these patients. Patients within this range are the subject of this study.

And it is not fair to make it clear that the exercises that apply are the same even if you have structured them on different intensities of stress.

There seems to be a big misunderstanding from the reviewer.

The main purpose of this study is to classify the exercise contents according to functional movement. It is not to apply the same exercise, but to select the exercise contents well within the functional movement classification considering the movement characteristics of stroke.As stated before, the purpose is not to uniformly apply standard program, but to systematically classify exercise selection. We do not offer standardized programs.The systematic classification (functional movement classification and ACSM's FITT) for the purpose of this study means that it is possible to configure various exercises according to the functional movement characteristics of various patients.

I do not agree with the statements in lines 219-227: Patients with stroke and cardiovascular disease do not do cardio training in the acute phase, even gradually. They have hemiplegia and are fighting for their lives, they are passively mobilized, and that is correctly and ethically done in the specialized neurorehabilitation wards. The cardio rehabilitation program set forth in lines 158-165 does not apply.

 Our researchers agree on your comment. The purpose of this study is to develop ICT-based exercise rehabilitation service contents. Classification was needed in order to create the program. Obviously, cardio training is not applied to acute-phase patients.Please understand once again that classification of all scopes is needed in developing the program.

I do not understand the terms: hematencephalon (in table 3, instead of hemorrhagic stroke I suppose, exercise rehabilitation service, and movement rehabilitation services, have nothing to do with medical rehabilitation.

 Thank you for your comment. As explained above, classification was made because cerebral infarction is divided as hematencephalon and cerebral infarction.

So, my final suggestion is to completely separate the two categories, extremely different as pathophysiology, clinic and of course medical approach, principles and techniques of rehabilitation!!!!!

 Thank you for your significant comments.Although stroke patients have the pathological specificity of neurological diseases, the community should consider providing public exercise rehabilitation services addition to medical care as long-term rehabilitation is required.This is the biggest purpose of the national R&D project in which the research team is currently participating.

From the prescribing point of view of exercise rehabilitation, in addition to sharing the data of pathological patients, it is important to know how to exercise which parts of the body. From our study, exercise specialists include exercise rehabilitation specialists, sports rehabilitation specialists, and physical therapists.

In order to offer effective exercise, exercise experts applied functional movement classification and ACSM’s FITT to systematize the purpose of exercise, which has been confirmed in various related prior studies.

In the preceding reference literature, there was no difference between stroke and musculoskeletal system in systematic classification of movement area, functional movement classification, applied movement type, frequency, and number of repetitions. Therefore, we proceeded to classify the two diseases in the same system.

In this ICT-based exercise rehabilitation service, exercise content is selected according to the purpose of the exercise, and it becomes possible to compose a variety of exercises according to the functional movement characteristics of various patients.Therefore, I hope that you understand the purpose of this study once again.